# The structure and dynamics of secretory component and its interactions with polymeric immunoglobulins

Beth M Stadtmueller[1], Kathryn E Huey-Tubman[1], Carlos J López[2,3†], Zhongyu Yang[2,3‡], Wayne L Hubbell[2,3], Pamela J Bjorkman[1*]

[1]Division of Biology and Biological Engineering, California Institute of Technology, Pasadena, United States; [2]Jules Stein Eye Institute, University of California, Los Angeles, United States; [3]Department of Chemistry and Biochemistry, University of California, Los Angeles, United States

*For correspondence: bjorkman@caltech.edu

Present address: † Grifols Biologicals Inc., Los Angeles, United States; ‡ Department of Chemistry and Biochemistry, North Dakota State University, Fargo, United States

**Competing interests:** The authors declare that no competing interests exist.

**Abstract** As a first-line vertebrate immune defense, the polymeric immunoglobulin receptor (pIgR) transports polymeric IgA and IgM across epithelia to mucosal secretions, where the cleaved ectodomain (secretory component; SC) becomes a component of secretory antibodies, or when unliganded, binds and excludes bacteria. Here we report the 2.6Å crystal structure of unliganded human SC (hSC) and comparisons with a 1.7Å structure of teleost fish SC (tSC), an early pIgR ancestor. The hSC structure comprises five immunoglobulin-like domains (D1-D5) arranged as a triangle, with an interface between ligand-binding domains D1 and D5. Electron paramagnetic resonance measurements confirmed the D1-D5 interface in solution and revealed that it breaks upon ligand binding. Together with binding studies of mutant and chimeric SCs, which revealed domain contributions to secretory antibody formation, these results provide detailed models for SC structure, address pIgR evolution, and demonstrate that SC uses multiple conformations to protect mammals from pathogens.

## Introduction

The mucosa is fundamental to vertebrate survival, forming an elaborate extracellular environment, in which the immune system mediates host interactions with commensal and pathogenic agents. The human mucosa protects ~400 m$^2$ of epithelial barriers in the gut, lungs, urogenital tract, and associated tissues such as mammary glands. Protection is conferred largely through the function of the polymeric Immunoglobulin receptor (pIgR), which transports and stabilizes secretory antibodies and also functions as an innate immune factor (*Kaetzel, 2005*). Human pIgR is a glycosylated type I membrane protein consisting of a 620-residue ectodomain with five tandem immunoglobulin-like (Ig-like) domains, a 23-residue transmembrane domain, and a 103-residue intracellular domain (*Hamburger et al., 2006*) (*Figure 1A*). pIgR is the oldest identifiable Fc receptor, first emerging in teleost (bony) fish. Throughout evolution, the number of Ig-like domains in the pIgR ectodomain increased; typically, bony fish express a two-domain variant, birds, amphibians and reptiles a four-domain variant, and mammals a five-domain variant (D1-D5) (*Akula et al., 2014*). Mammals, including rabbits and cows, express an alternatively-spliced variant containing D1, D4 and D5 (*Deitcher and Mostov, 1986*; *Kulseth et al., 1995*).

The pIgR is expressed on the basolateral surface of epithelial cells where the ectodomain binds polymeric forms of IgA and IgM produced by local plasma cells. Similar to other antibody classes, each IgA and IgM monomer contains two Fab arms and one Fc region (a dimer of IgA heavy chain domains Cα2 and Cα3 or IgM domains Cμ2, Cμ3 and Cμ4). Unlike other antibody classes, IgA and

**eLife digest** A sticky substance called mucus lines our airways and gut, where it acts as a physical barrier to prevent bacteria and other microbes from entering the body. Mucus also contains proteins called antibodies that can bind to and neutralize molecules from microbes (known as antigens).

The primary antibody found in mucus is called Immunoglobulin A. This antibody is produced by immune cells within the body and must pass through the "epithelial" cells that line the airway or gut to reach the layer of mucus. These epithelial cells have a receptor protein called the polymeric immunoglobulin receptor (pIgR) that binds to Immunoglobulin A molecules, transports them across the cell, and then releases them into the mucus layer. The pIgR also releases Immunoglobulin A into breast milk, which protects nursing infants until their own immune system has developed.

When released into the mucus layer, the Immunoglobulin A antibodies remain attached to a portion of pIgR known as the secretory component. This part of the receptor serves to stabilize and protect the antibodies from being degraded and helps the antibodies to bind to other host and bacterial proteins. Researchers have noted that the secretory component can be released into the mucus even when it is not attached to an antibody. These "free" secretory components have been shown to help prevent bacteria and the toxins they produce from entering the body. Despite the importance of secretory component in immune responses, the three-dimensional structure of the secretory component and how it interacts with antibodies and bacteria remained unknown.

Here, Stadtmueller et al. use a technique called X-ray crystallography to determine a three-dimensional model of the free form of a secretory component from humans, and compare it to an ancestral secretory component protein found in fish. Further experiments on the human protein revealed how the structure of the secretory component changes when antibodies bind to it. Stadtmueller et al. propose a model for how both forms of the secretory component can protect the body from microbes and other external agents. The next challenge is to develop a three-dimensional model of the secretory component when it is bound to Immunoglobulin A.

IgM monomers can polymerize, producing dimeric IgA (dIgA), pentameric IgM (pIgM), and to a lesser extent, higher order polymers. Adjoining monomers are linked tail-to-tail through heavy chain C-terminal extensions (tailpiece; tp) and a 137-residue protein called joining (J) chain and are stabilized through disulfide bonds (*Hamburger et al., 2006*). The transport cycle of pIgR (*Figure 1B*) starts when pIgR at the basolateral membrane binds dIgA or pIgM, and the resulting complex is transcytosed to the apical surface where proteases cleave the pIgR ectodomain (now referred to as secretory component, SC), releasing it into the mucosa as a complex with dIgA or pIgM. These complexes, Secretory IgA (SIgA; the predominant mucosal antibody) and Secretory IgM (SIgM), exclude pathogens from the epithelial barrier and promote host relationships with commensal bacteria through innate and adaptive mechanisms. SC specifically protects secretory Ig from proteolytic degradation and confers innate recognition functions upon SIgA and SIgM, allowing them to bind and exclude bacteria (*Kaetzel, 2005*). Although binding to polymeric Ig stimulates pIgR transcytosis (*Song et al., 1994*), up to 50% of pIgR in humans trafficks to the apical surface and is released as unliganded, or free, SC (*Brandtzaeg, 1971*), which can exclude pathogenic bacteria and bacterial toxins through protein-protein or protein-glycan interactions (*Kaetzel, 2005*). In humans, SC binds the major *Streptococcus pneumoniae* surface protein, choline binding protein A (CbpA), using protein-protein interactions, and to pathogenic bacteria such as *H. pylori, E. coli* and *Shigella* spp, using complex carbohydrates attached to one or more of its seven potential *N*-linked glycosylation sites (*Kaetzel, 2005*). SC carbohydrates have also been shown to facilitate binding to host mucus, host protein IL-8, commensal bacteria *Lactobacillus* and *Bifidobacteria* and contribute to microbiota homeostasis of the intestinal mucosa (*Kaetzel, 2005*; *Mathias and Corthesy, 2011*). SC and SIgA interactions with pathogens and commensals are thought to be especially important for nursing infants, who ingest large quantities of maternal free SC and SIgA (*Hurley and Theil, 2011*; *Rogier et al., 2014*).

Understanding the structure(s) of the pIgR ectodomain (hereafter called SC) and how it interacts with ligands and pathogens is of interest because its critical role in immunity requires the protein to accommodate binding, transport and protection of secretory antibodies while also conferring innate protection in both free and liganded forms. High-resolution structural information for SC and the SC interactions with polymeric immunoglobulin (pIg) ligands is limited to a crystal structure of the human SC D1 domain, which adopts an Ig-variable (V)-like fold (*Hamburger et al., 2004*). The structures and contributions of D2-D5 to intact SC function are largely unknown. D1 is both necessary and sufficient for binding to pIg Fcs and is also thought to interact with J-chain because pIgR transports only J-chain–containing pIgs, and isolated D1 does not bind monomeric IgA. D1 binding to pIg is partly mediated by three D1 loops that are structurally equivalent to the antigen-binding complementarity determining regions (CDRs) of immunoglobulin variable domains (*Hamburger et al., 2006*). Binding to dIgA can be further stabilized by a disulfide bond between SC D5 and Fcα Cα2; however, this interaction is absent in some SIgA complexes and does not form in SIgM (*Almogren et al., 2007*; *Hamburger et al., 2006*).

Here we report the first crystal structures of intact SC proteins, comparing the highly-evolved five-domain human SC (hSC) and a two-domain teleost fish SC (tSC), a relative of the first vertebrate SC ancestor. We characterized the conformation and dynamics of free and liganded hSC in solution, and used structure-based alignments to create mutant and chimeric SCs to determine how individual domains contribute to ligand binding. These results provide a detailed model for SC structure and pIg binding mechanisms, demonstrating that mammalian SC evolved to adopt a compact, closed triangular structure, which opens upon ligand binding, whereas two-domain SC ancestors consist of tandem domains arranged in an elongated conformation. For hSC, we show that each of the five domains adopt distinct associations with each other in unliganded versus liganded forms, and that each contributes uniquely to dIgA and pIgM recognition and secretory antibody formation.

## Results

### Crystal structure of hSC

The crystal structure of hSC (*Figure 1C*) was determined to 2.6Å resolution ($R_{cryst}$ = 20.1%; $R_{free}$ = 25.4%) (*Supplementary file 1*). The final model (540 ordered residues of 549 total) revealed five Ig-like domains (D1-D5) arranged into a compact triangle (three sides of ~70Å, ~70Å and ~90Å) in which D2-D3 and D4-D5 form two of the sides, and D1 contacts both D2 and D4-D5 to form the third side (*Figure 1C*). The domains lie in a plane such that the triangle thickness is roughly equal to that of a single domain (~40Å) (*Figure 1C,D*). The overall arrangement involves extensive interfaces between all five domains and a small solvent-accessible hole (~14Å diameter) in the center. As defined in *Figure 1D*, the hSC front face shows all five domains. A 90° clockwise rotation reveals a side face dominated by D2 and D3; another 90° clockwise rotation reveals the back face showing all five domains, and a further 90° rotation reveals a side face comprising D4 and D5. A fifth face is formed at the bottom of the hSC triangle (90° from the front and back faces), which includes D5, D1 and D2, and all domains are visible when viewed from the top. Important SC motifs, including CDRs, some residues implicated in ligand binding, and potential *N*-linked glycosylation sites, are largely solvent exposed and distributed on all faces of the molecule (*Figure 1C,D*). The D1 CDR1 and CDR3 loops, which contribute to dIgA and pIgM binding (*Kaetzel, 2005*), are exposed exclusively on the front face, whereas the D4 CDRs are exposed only on the back face. The D2, D3 and D5 CDRs are exposed on both faces and at least one side, and D5 residues Cys468 and Cys502, which can disulfide bond with the dIgA residue Cys311 (*Hamburger et al., 2006*), are exposed on the D4-D5 side.

The D1, D2, D4, and D5 domains of hSC include seven potential *N*-linked glycosylation sites that anchor carbohydrates involved in hSC interactions with bacterial and host lectins (*Kaetzel, 2005*). We observed partially ordered glycans at four sites (D1 sites Asn65 and Asn72, D2 site Asn168, and D5 site Asn481), which do not contact protein portions of hSC. These observations suggest that carbohydrates are unlikely to stabilize the hSC domain arrangement directly, but are optimally positioned to allow hSC to bind pIg ligands and to facilitate lectin interactions with free hSC and SIgA. At least one *N*-linked glycosylation site is visible from each orientation; however, six of the seven potential glycosylation sites are clustered on the back and bottom faces of the molecule (*Figure 1C, D*).

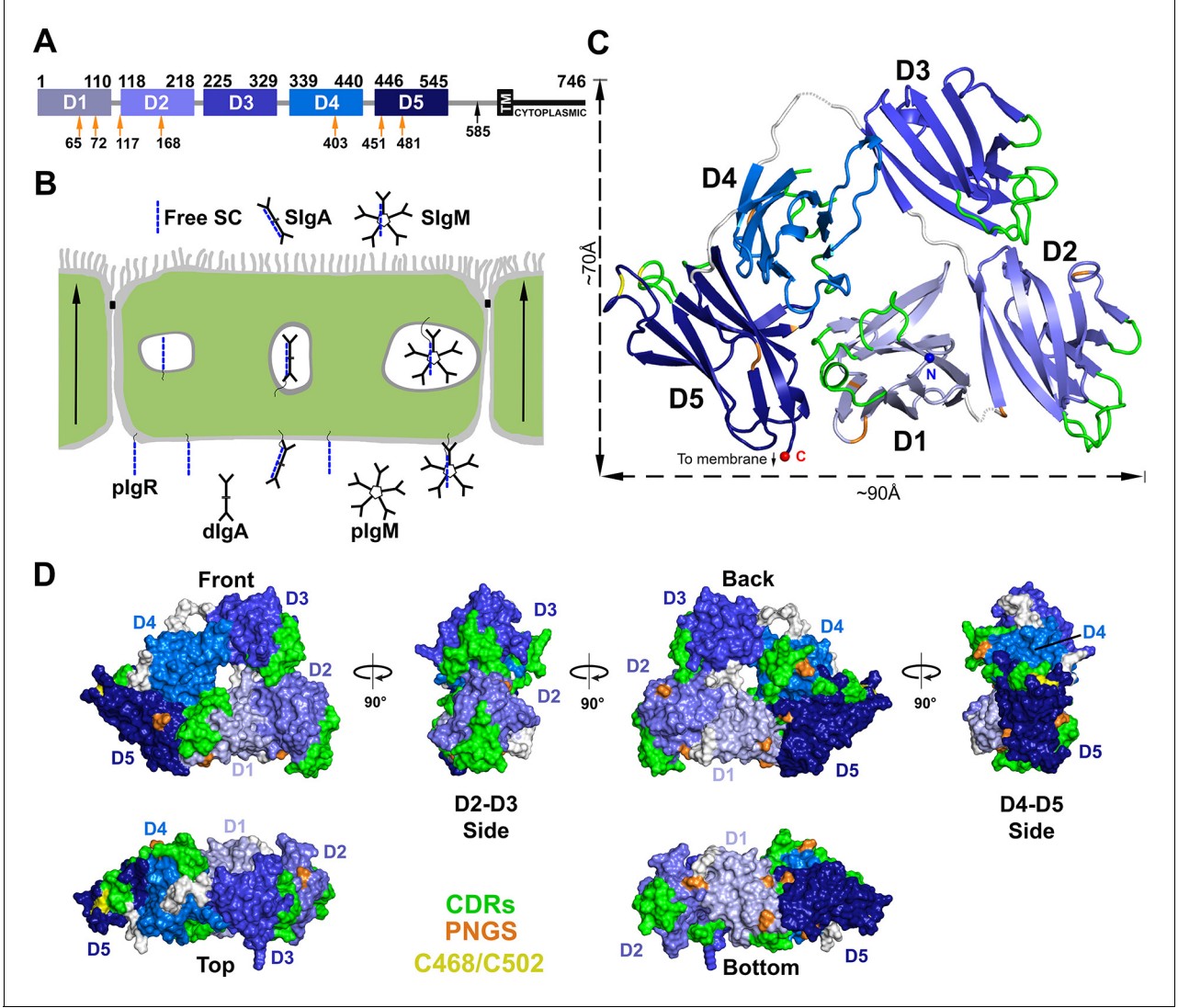

**Figure 1.** Structure of hSC. (A) Schematic of mature human pIgR protein indicating Ig-domain (D1-D5) boundaries. The proteolytic cut site that releases hSC from the apical membrane (black arrow), the 23-residue transmembrane region (TM), cytoplasmic tail, and potential *N*-linked glycosylation sites (PNGS, orange arrows) are indicated. (B) Schematic epithelial cell layer showing basolateral to apical transcytosis (arrows) of pIgR and release of free SC, SIgA, and SIgM. (C) Cartoon representation of the hSC structure viewed from the front face colored to highlight CDR loops (green), D5 Cys468 and Cys502 (yellow), PNGS (orange), domain linkers (grey), and hSC termini (N-terminus: blue sphere; C-terminus: red sphere). (D) Molecular surface representation of the hSC structure shown in six orientations and colored as in (C).

## Folding topologies of hSC domains

Consistent with predictions (*Mostov et al., 1984*), each hSC domain adopts an Ig superfamily (IgSF) fold. IgSF domains share a basic folding topology consisting of two β-sheets linked through a con-served disulfide bond. The domains are classified as Variable (V), Intermediate (I) or Constant (C) (*Wang, 2013*). D1-D4 each adopt an Ig-V-like fold, which typically contain β-strands A, B, E, D on one sheet and strands A', G, F, C, C', C" on the other. As reported for D1, the loops connecting strands B-C, C'-C", and F-G are similar to IgV CDRs (*Hamburger et al., 2004*). In contrast to D1, domains D2-D4 each lack an A strand that would normally pair with the β-sheet containing strands B, E, and D (*Figure 2A–D*). The A strand-equivalent residues in these domains instead form the D1-D2, D2-D3 and D3-D4 linkers. In addition to the canonical Ig disulfide bond connecting strands B and F, domains D1, D3, D4 and D5 include a second disulfide that links the C and C' strands. Despite lacking the C-C' disulfide, the relative orientation of the C and C' β-strands in D2 is similar

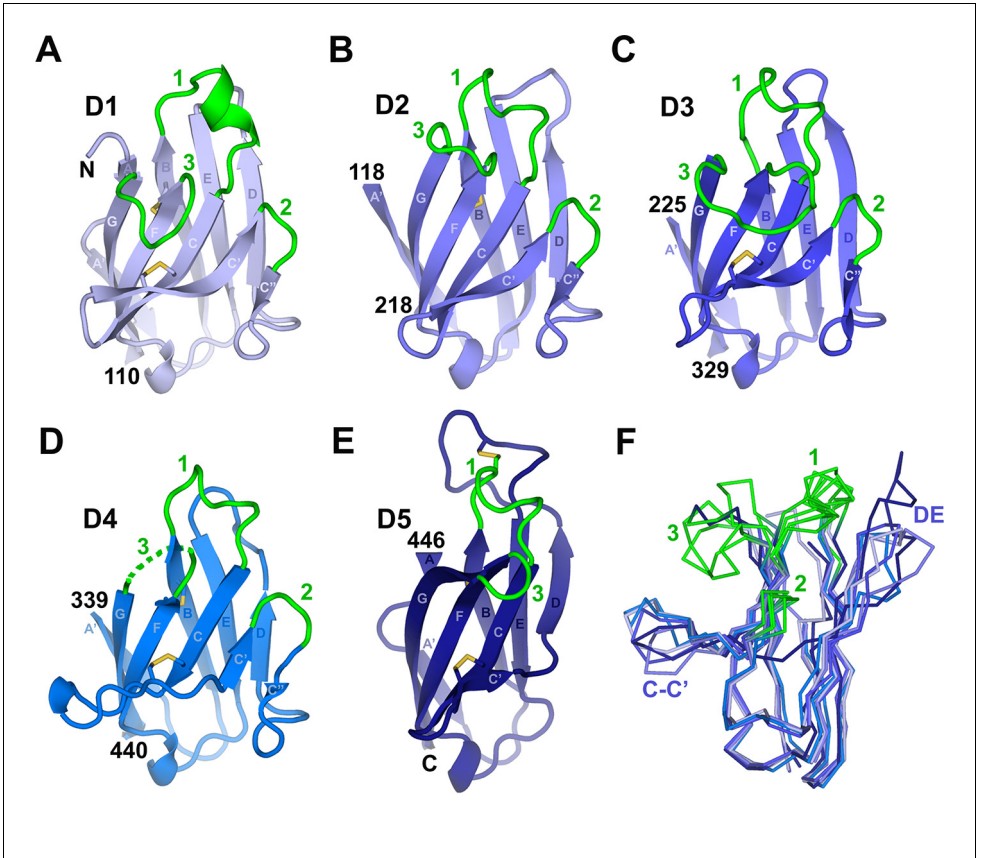

**Figure 2.** Individual hSC domains. (**A-E**) Cartoon representations of hSC domains shown in the same orientation. Disulfides are shown as sticks (yellow), and CDR loops (green), N- and C-terminal residues, and β-strands within Ig domain topology are labeled. The D4 CDR3 is likely flexible because five of its seven residues are disordered (dashed lines). (**F**) Ribbon diagram showing Cα traces of aligned domains D1-D5 viewed from an orientation ~90° clockwise from (**A-E**) with CDR loops and regions with structural differences among domains indicated. See also *Figure 2—figure supplement 1*.

The following figure supplement is available for figure 2:

**Figure supplement 1.** Comparison between human D1 and D3 CDR3 loop structures.

to other SC domains (*Figure 2*). D5 topology diverges from the canonical IgV fold because, in common with both the IgI and IgC folds, it lacks the C″ strand and associated CDR2 loop (*Figure 2E*). D5 also diverges from D2-D4 because residues following the D4-D5 linker hydrogen bond to the B strand, providing D5 with a two-residue A strand, a feature found in IgV and IgI but absent in IgC (*Wang, 2013*). Taken together, D5 topology resembles IgI, although it lacks conserved interactions that typically link loops equivalent to CDR1 and CDR3 (*Wang, 2013*).

The CDR-equivalent loops in hSC D1-D5 display differences that suggest distinct functional roles. We previously reported that D1 CDR1 includes an α-helical turn formed in part by residues implicated in ligand binding (*Hamburger et al., 2004*). Here we find that the CDR1 loops from all hSC domains contain a helical turn capped by a conserved lysine residue. However, the D2-D5 CDR1 loops include $3_{10}$-helices instead of the CDR1 α-helix in D1 (*Figure 2*). The D5 CDR1 differs from the other CDR1s because it is disulfide bonded to the neighboring DE loop (residues Cys468 in CDR1 and Cys502 in the DE loop) (*Figure 2E*). Notably, the D5 DE loop is 12 residues long, compared to 3-4 residues in D1-D3 and 7 residues in D4, and extends ~10Å beyond the position occupied by DE loops in other domains (*Figure 2E,F*). Despite the disulfide bond to the D5 CDR1, side chain resolution was poor and the atomic B-factors were high for several residues in the DE loop, suggesting flexibility. CDR3 also exhibited divergence in conformation and length among D1-D5, varying from

two residues in D5 to ten residues in D3, although D1 and D3 CDR3s occupy similar positions (*Figure 2*, *Figure 2—figure supplement 1*). The two-residue CDR2s adopt similar conformations among D1-D4.

## hSC domain orientations and interfaces

The compact, triangular arrangement of hSC domains is distinct from tandem domains in IgSF proteins, such as CD4 (*Wu et al., 1997*), which are elongated with nearly co-linear domains. Instead, adjacent hSC domains are not co-linear and share distinct interfaces (*Figure 3A*, *Figure 3—figure supplements 1, 2*). D1 and D2 are linked by a partially disordered and likely flexible linker and are related by an ~82° inter-domain angle. Their interface comprises residues in the D1 A', G, F, C, and C' strands, which contact residues along the D2 A' strand and the D2-D3 linker (1,156 Å² total buried surface area) (*Figure 3A*, *Figure 3—figure supplement 2A*). Domains D2 and D3 (related by ~152°) are nearly co-linear and connected by a potentially rigid four-residue linker with two conserved prolines. Residues in the D2 A'-B loop, B strand, and the E-F loop contact residues in the D3 CDR3 and CDR1 loops, forming an interface that buries 790 Å² total surface area (*Figure 3A*, *Figure 3—figure supplement 2B*). D3 connects to D4 via a partially disordered and likely flexible seven-residue linker. The two domains are related by ~118° and share a small interface (596 Å² total buried surface area) involving residues in the D3 A' and B strands and the D4 C-C' loop. D4 and D5 are related by ~96° and form an extensive interface (1520 Å² total buried surface area), in which residues in the D4 A'-B loop and B strand, CDR2 and C"-D loops, and D and E strands contact residues in the D5 G and F strands, C-C' loop and CDR3 (*Figure 3*, *Figure 3—figure supplement 2C*), stabilizing the interface with numerous hydrogen bonds and hydrophobic interactions.

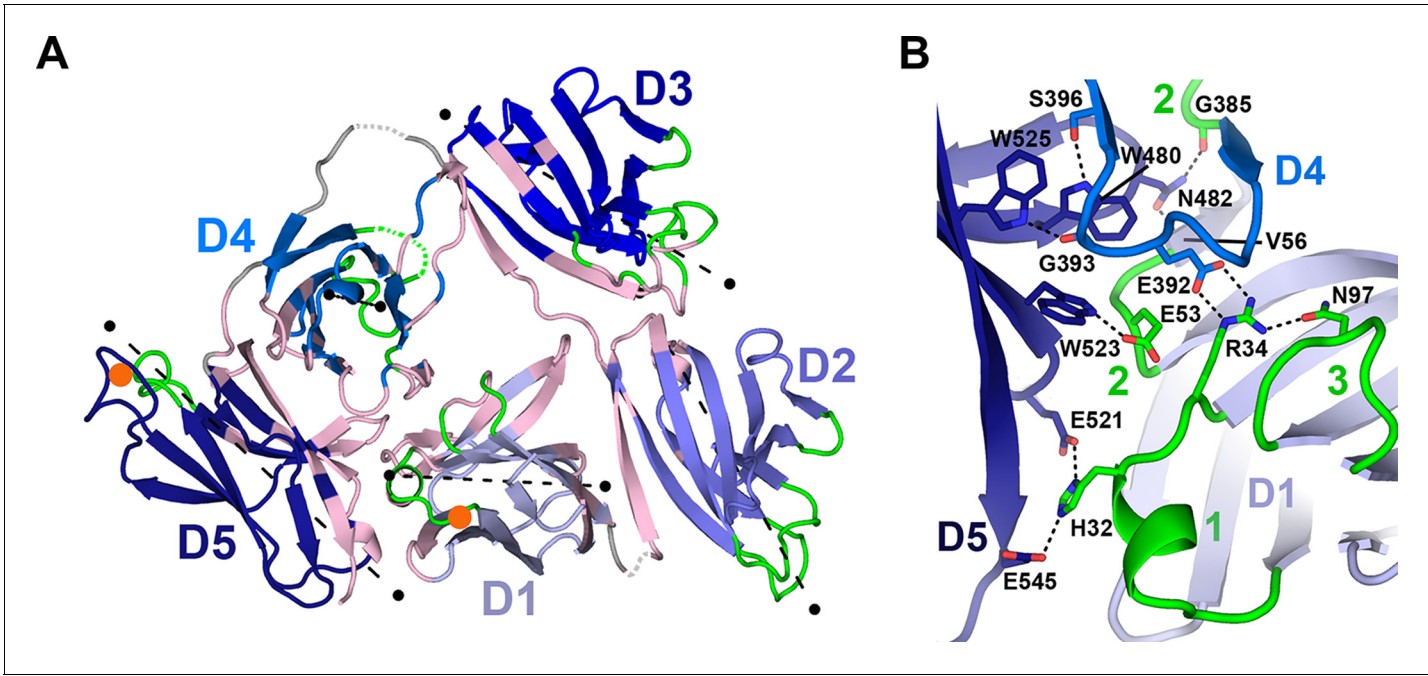

**Figure 3.** Domain interfaces of hSC. (**A**) Cartoon representation of the hSC structure showing the front face view with interface residues colored pink and axes used to determine angles between domains shown as dashed lines (each line is 50Å long). The approximate position of D1 CDR1 Pro26 and D5 CDR1 Cys468 are shown as orange circles. (**B**) Cartoon representation of the D1-D4-D5 interface with residues involved in putative hydrogen-bonding interactions shown as sticks and including three conserved D4-D5 interface residues, Trp480, Trp523 and Trp525, which bracket the D4 C"-D loop, and are positioned within hydrogen bonding distance of D4 Ser396 and Gly393. See also *Figure 3—figure supplements 1, 2*.

The following figure supplements are available for figure 3:

**Figure supplement 1.** Sequence alignment of representative SC D1-D5 domains.

**Figure supplement 2.** Domain interfaces.

An unexpected and likely functionally significant feature of the hSC structure is the large interface (1480 Å² total buried surface area) formed between domains D1-D4-D5 (*Figure 3*). D1 and D4 are approximately anti-parallel, with CDR loops pointing in opposite orientations and contacts formed by the C″ strands and neighboring residues from both domains. D5 is almost perpendicular to D1 and D4 (related by ~96° and ~101°, respectively), with its CDR loops pointing away from D1, separating ligand-binding motifs in D1 and D5 CDR1 loops by nearly 45Å (as measured between D1 CDR1 Pro26 and D5 CDR1 Cys468, orange circles, *Figure 3A*). Notably, the interface includes residues from all three of the D1 CDR loops and buries CDR2 at the bottom of a depression formed by the three domains.

Important contacts between D4 and D1 (*Figure 3B*) that stabilize the triangular shape of hSC include a salt bridge between D4 Glu392 and D1 Arg34, the last residue in D1 CDR1. Arg34, which is critical for pIgR binding to dIgA (*Coyne et al., 1994*), also hydrogen bonds to D1 CDR3 residue Asn97, which stabilizes the position of the D1 CDR3 loop (*Hamburger et al., 2004*) to create a network of interactions between D4 and CDRs 1 and 3 in D1. This interface is further stabilized through interactions with the D5 C-C′ loop and flanking strands that protrude into the D1-D4 interface where D5 residues share contacts with D4 and D1, thereby bridging the three domains (*Figure 3B*). The keystone in this interaction is Asn482, whose side chain is located in the center of the C-C′ loop where it forms hydrogen bonds with the main chain oxygen of D4 Gly385 and the main chain nitrogen of D1 Val56. Similarly, Glu53, the residue at the center of the D1 CDR2, is buried by and within hydrogen bonding distance of D5 Trp523 (F strand), which also contacts D4. A set of pH-dependent stabilizing salt bridges occur between D1 CDR1 residue His32 and D5 residues Glu521 and Glu545, which are located in the D5 E-F loop and G strand, respectively (*Figure 3B*). His32, Arg34, Glu392, Asn482, and Glu521 are conserved in mammalian SC sequences, and Arg34 and Glu392 are also conserved in reptilian, amphibian, and avian SCs (*Figure 3—figure supplement 1*). It is notable that conserved D1 residues His32 and Arg34, as well as residues in D1 CDR2, are important or necessary for binding to dIgA and pIgM (*Coyne et al., 1994*; *Roe et al., 1999*), yet also stabilize the D1-D4-D5 interface and are largely inaccessible, while other putative ligand-binding residues in CDR1 and CDR3 (e.g., Tyr24-Asn30, Arg99-Lue101) remain exposed.

## Evaluating hSC conformational flexibility with DEER spectroscopy

The existence of flexible linkers, and in some cases, the absence of extensive stabilizing contacts between hSC domains, suggested that hSC might adopt a range of conformations. To evaluate the conformational flexibility of hSC in solution, we conducted double electron-electron resonance (DEER) spectroscopy on free and liganded forms of hSC variants containing pairs of nitroxide spin labels. A DEER experiment measures the probability distribution of inter-nitroxide distances in the 17–80Å range (*Jeschke and Polyhach, 2007*). The most probable distance and width of the distribution provide direct information on the structure and structural heterogeneity, respectively, and are presumed to reflect the amplitude of molecular motion in solution under physiological conditions. Compared to methods such as crystallography and electron microscopy that capture snapshots of protein structure, DEER has the advantage of providing analytical data that describe a protein's structural heterogeneity and flexibility (*Hubbell et al., 2013*).

Guided by the hSC structure, we prepared spin-labeled variants, D1-67R1/D5-455R1, D1-67R1/D5-491R1, and D1-80R1/D5-491R1, containing the R1 nitroxide side chain at the indicated positions (*Figure 4A,B*), to monitor changes in spatial proximity between D1 and D5. An intra-domain pair D5-455R1/D5-491R1 was prepared to monitor domain flexibility. All spin-labeled proteins bound dIgA and pIgM with kinetics indistinguishable from unlabeled hSC showing that spin labeling did not disrupt ligand binding (*Figure 4—figure supplement 1A*).

DEER time domain data and resulting distance distributions for all R1 pairs in the unliganded protein demonstrated most probable distances within 1Å of the expected distances based on the hSC crystal structure (*Figure 4C–F*, *Figure 4—figure supplement 1B*), indicating that the crystal structure represents the predominant solution structure of hSC at the D1-D5 interface. The intra-domain distance distribution for D5-455R1/D5-491R1 showed one dominant peak at 25Å with a narrow width at half-height of 3.5Å, characteristic of R1 pairs in a rigid protein structure (*Lerch et al., 2014*). By contrast, distance distributions for the inter-domain pairs were multimodal and broader, suggesting contributions from protein flexibility. The flexibility could involve local motions of secondary structure elements containing the spin labels and/or subtle inter-domain motions between D1

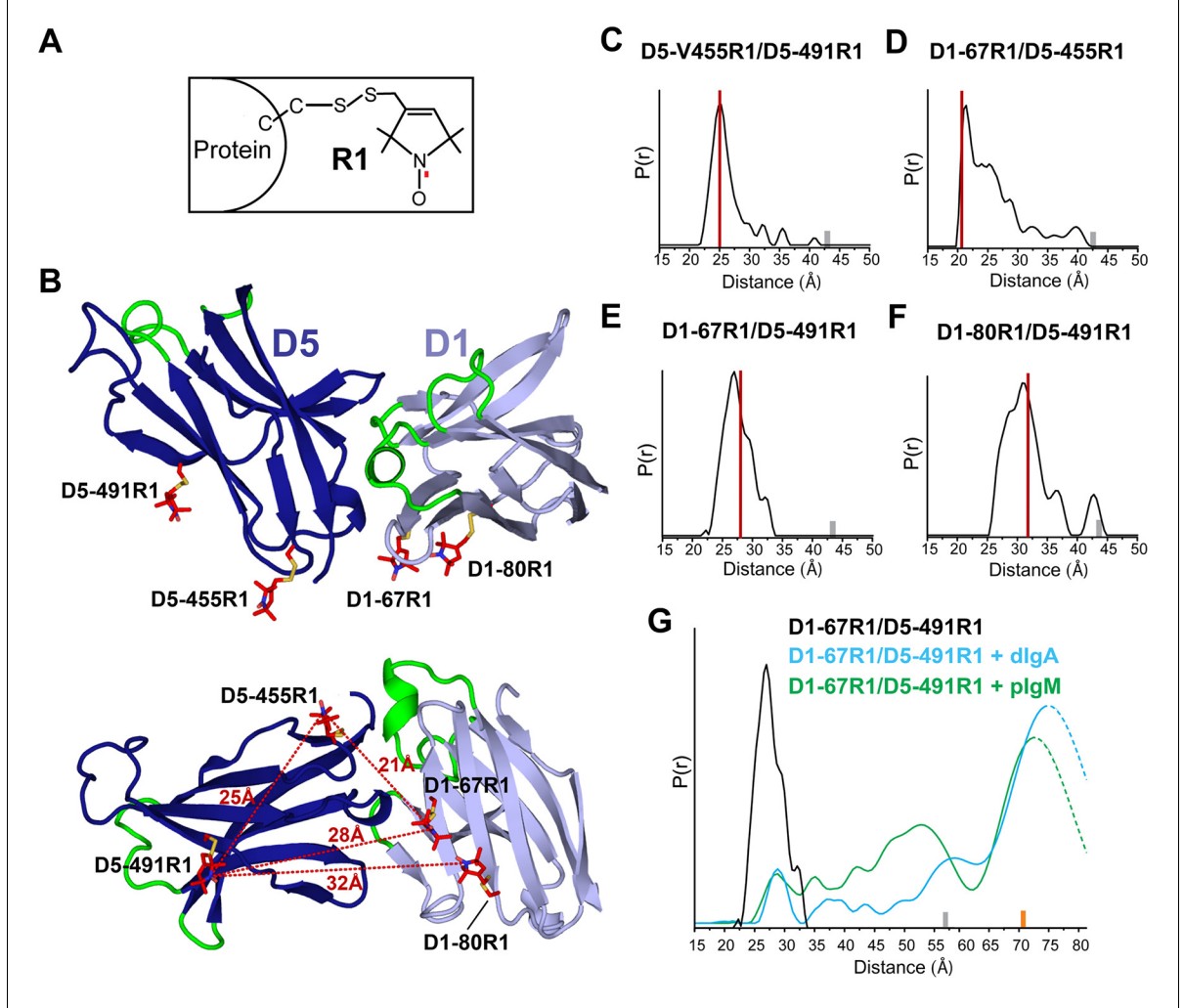

**Figure 4.** DEER spectroscopy of spin-labeled hSC. (A) Structure of R1 nitroxide (unpaired electron shown in red) attached to a cysteine residue. (B) Cartoon representation in two orientations showing modeling of an R1 side chain (red sticks) at the indicated sites on the hSC crystal structure. The distances measured are indicated as red dashed lines in the lower panel. (C-F) Distance distributions for the indicated mutants obtained after model-free fitting of the dipolar evolution function (DEF). The vertical red lines indicate the expected interspin distance based on modeling of the R1 side chain on the hSC crystal structure. (G) Same as C-F but for D1-67R1/D5-491R1 complexes with dIgA (cyan) or pIgM (green). Minor peaks at 27Å likely correspond to unliganded SC, and peaks observed with maxima at ~57Å upon dIgA binding and at ~53Å upon pIgM binding may reflect structural heterogeneity in ligand structure or arise from intermediate binding states. The relative populations of these states cannot be determined due to existence of distance probabilities outside the reliable range of detection. The gray and orange bars indicate the upper limit of reliable distance and shape of the distribution for unliganded and liganded hSC variants, respectively (see Experimental Procedures). For D1-67R1/D5-491R1 complexes, distances above 70 Å are beyond detection limits (dashed traces). See also *Figure 4—figure supplement 1*.

The following figure supplement is available for figure 4:

**Figure supplement 1.** SPR data, time domain data and CW EPR spectra and control DEER measurements.

and D5, but the relatively small amplitude of these motions (<10Å) indicated that D1 and D5 are in contact.

To investigate whether ligand binding induces structural changes in the hSC D1-D5 interface, and to obtain structural data describing SIgA and SIgM complexes, we measured the interspin distance between D1-67R1 and D5-491R1 in the presence of dIgA and pIgM. As shown in *Figure 4G* and *Figure 4—figure supplement 1C*, ligand binding induced a dramatic separation of D1 and D5, leading to a broad distance distribution between the R1 residues, extending beyond 70Å. The position and widths of the distance distribution for distances beyond 70Å are not well-determined, but the data

are consistent with movements of more than ~42Å between D1-D5. Despite this remarkable domain separation, the intra-domain distance distribution of D5-455R1/D5-491R1 remained unchanged (*Figure 4—figure supplement 1D*), and the continuous wavelength (CW) EPR spectra did not change for any of the sites (*Figure 4—figure supplement 1E*), indicating that the local structure around R1 was unperturbed. The broad distance distributions we observed for hSC in the hSC:dIgA and hSC:pIgM complexes suggest that hSC binding progresses through intermediate states and/or that liganded hSC is highly flexible. These observations further suggest that SIgA and SIgM structures are richly heterogeneous.

## Structure of teleost fish SC

To compare SC structures across evolution and to facilitate the design of chimeric proteins for characterizing SC-ligand interactions, we solved the 1.7Å crystal structure of teleost fish SC (tSC) ($R_{cryst}$ = 18.5%; $R_{free}$ = 21.5%) (*Supplementary file 1*). Teleost fish express the oldest recognizable pIgR protein (*Akula et al., 2014*), which transports polymeric versions of IgM and IgT/IgZ (a teleost Ig specialized in mucosal immunity) to the mucosa (*Sunyer, 2013*). The fish genome does not encode a J-chain, thus fish pIgs are thought to be structurally divergent from their human counterparts (*Flajnik, 2010*). In addition, tSC contains just two extracellular domains, tD1 and tD2. tD1 and tD2 share homology with mammalian D1 and D5, respectively, but CDR loop residues implicated in mammalian SC interactions with pIg are not conserved (*Feng et al., 2009*; *Hamuro et al., 2007*; *Rombout et al., 2008*).

The structure of tSC revealed two tandem Ig-like domains related by ~90° (*Figure 5A*). Like hSC domains D1 and D3-D5, tSC domains are IgSF folds that include the canonical B to F strand disulfide bond and a second disulfide bond linking the C and C' strands. tD1 and tD2 share a similar topology (*Figure 5—figure supplement 1A,B*), but only 72 of 97 Cα atoms could be aligned (rmsd = 0.87Å). Aligning fish and human SC domains resulted in rmsds of 0.77Å for tD1 and D1 (83 of 100 Cα atoms) and 1.08Å for tD2 and D5 (64 of 97 Cα atoms) (*Figure 5B–D*). The largest structural differences between tSC and hSC domains are in regions between the C' and D strands, where tD1 residues at positions similar to the C" strand are disordered and the tD2 C" strand hydrogen bonds to the D strand on the A-B-E-D face rather than to the C' strand on the A'-G-F-C face (*Figure 5*, *Figure 5—figure supplement 1A,B*), similar to other IgSF proteins such as IREMs (immune receptors expressed on myeloid cells) (*Marquez et al., 2007*). These differences are coupled with distinct features in tSC CDR2 loops and neighboring residues, which include a patch of negative electrostatic potential on tD2 formed from six negatively charged residues in the CDR2, C", and D strands. The tSC CDR1 and CDR3 loops also adopt distinct conformations. In contrast to the human D1 CDR1 α-helical turn, both tSC CDR1 loops include a $3_{10}$ helical turn and are structurally similar to human D5 CDR1. The tD1 CDR1 is solvent exposed and highly conserved among representative fish SC sequences, while the tD2 CDR1 is partly in contact with tD1 and contains fewer conserved residues (*Figure 5A*, *Figure 5—figure supplement 1C,D*). The residues in the CDR3 loops are largely conserved among fish (*Figure 5—figure supplement 1C*), but differ between tSC domains; the tD1 CDR3 is elongated, extending toward CDR1 and containing a short helical turn, whereas the tD2 CDR3 is shorter, similar to the D5 CDR3 (*Figure 5A–C*). tD2 has a 4-residue DE loop, similar to counterpart loops in human D1-D4 and in contrast to the extended loop and disulfide link to CDR1 found in human D5. The interface between the two tSC domains is formed by residues in the tD1 A'-B and E-F loops and residues in tD2 CDR1 and DE loop, burying a total surface area of 591Å$^2$. Ten of the 17 residues in the interface are well conserved among representative fish species, and five inter-domain hydrogen bonds may limit inter-domain flexibility (*Figure 5—figure supplement 1C,D*). Collectively, structural differences between the tD1-tD2 and their human D1 and D5 counterparts may represent adaptations that are advantageous for binding to species-specific ligands.

## SC-pIg interactions

Differences between the fish and human SC structures, together with data demonstrating that hSC undergoes a conformational change upon pIg binding, motivated us to investigate how each domain contributes to ligand binding. Our approach involved evaluating the ability of short hSC variants and human-fish SC chimeric proteins to bind human dIgA (*Figure 6*) and pIgM (*Figure 6—figure supplement 1*). Unless otherwise noted, hSC D5 residues Cys468 and Cys502 were mutated to alanine to

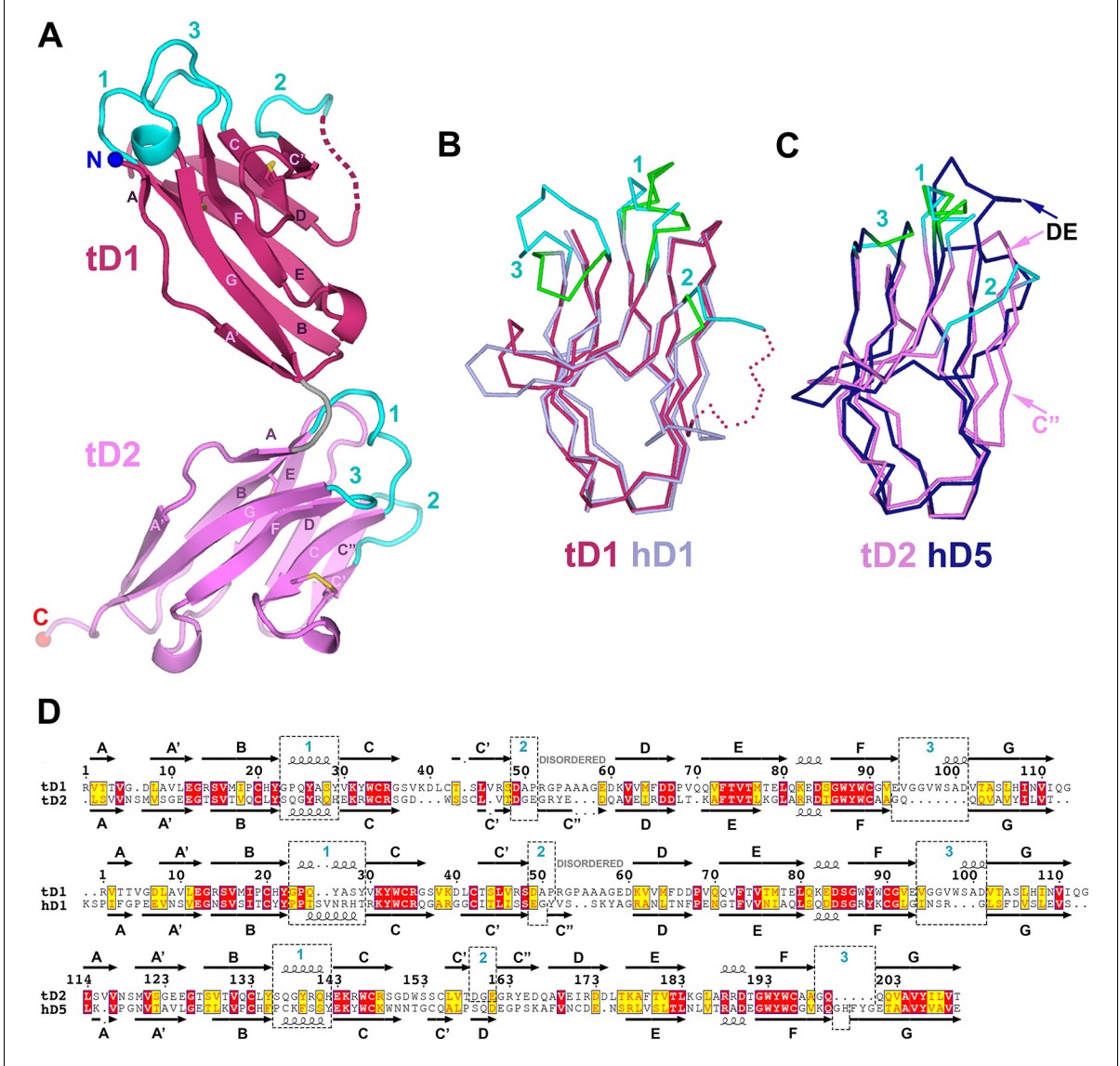

**Figure 5.** Structure of tSC. (A) Cartoon representation of the tSC structure with disulfides shown as yellow sticks, CDR loops colored cyan, N- and C-termini indicated (blue and red spheres) and Ig domain topology labeled. (B) Ribbon diagram showing Cα traces of tD1 and human D1 (hD1) following alignment. tD1 is deep salmon with cyan CDRs, and hD1 is light purple with green CDRs. (C) Ribbon diagram showing aligned Cα traces of tD2 and human D5 (hD5). tD2 is pink with cyan CDRs, and hD5 is dark blue with green CDRs. (D) Structure-based sequence alignments of tD1-tD2, tD1-hD1, and tD2-hD5 with corresponding secondary structure and CDR boundaries shown. See also *Figure 5—figure supplement 1*.

The following figure supplement is available for figure 5:

**Figure supplement 1.** tSC domain interfaces and sequence conservation.

prevent covalent binding to dIgA. All hSC variants and human-fish chimeric proteins were monomeric and monodisperse as verified by size exclusion chromatography (SEC) and/or SEC with in-line multi-angle light scattering (MALS) (data not shown). Although the stoichiometry of hSC to pIg in SIg complexes is reportedly 1:1, binding is thought to occur through a multi-step mechanism (*Hamburger et al., 2006*). Consistent with this model and the demonstration of a ligand binding-induced conformational change in hSC, the binding profiles of our SC variants failed to fit single state kinetic models. Because uncertainties in the binding mechanism precluded accurate selection

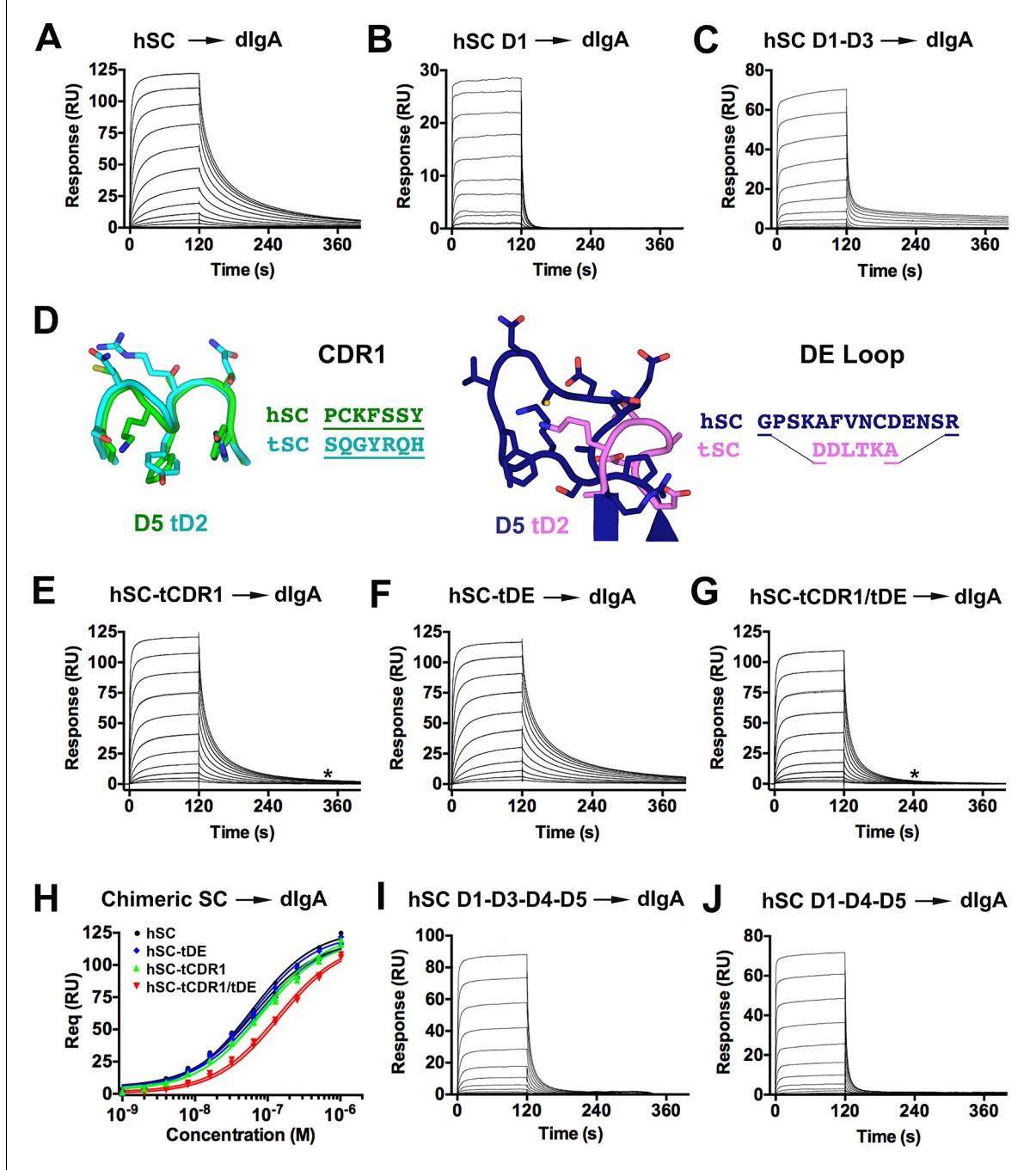

**Figure 6.** hSC and chimeric SC binding to dIgA. (A-C) Sensorgrams showing hSC, D1, and D1-D3 binding to dIgA. (D) Cartoon representations and associated sequences of CDR1 and DE loop residues substituted in chimeric proteins following structural alignments and modeling. (E-G) Sensorgrams for interactions of chimeric SC proteins and dIgA. The time at which complete dissociation occurs is indicated (astrisk). (H) Equilibrium binding response versus the log of concentration for the sensorgrams in (A, E-G) and replicate experiments (not shown). Average $K_D$ values for two replicates for each SC variant were: hSC (63+/-4nM), hSC tCDR1 (82+/-1nM), hSC tDE (65+/-3nM), hSC tCDR1/tDE (140+/-7nM). (I-J) Sensorgrams showing hSC D1-D3-D4-D5 and D1-D4-D5 binding to dIgA. See also *Figure 6—figure supplement 1*.

The following figure supplement is available for figure 6:

**Figure supplement 1.** hSC and chimeric SC binding to pIgM.

of a kinetic model, we qualitatively monitored differences in the SPR binding profiles to identify changes in kinetics among protein variants.

hSC D1-D5 exhibited concentration-dependent binding to dIgA with relatively slow association and dissociation phases (*Figure 6A*). In contrast, and consistent with previously published SPR experiments (*Hamburger et al., 2004*), D1 association and dissociation were rapid (*Figure 6B*), demonstrating that D2-D5 contribute to the dIgA binding mechanism even in the absence of covalent interactions mediated by D5 Cys468/Cys502. To investigate potential contributions from other hSC domains, we tested binding of D1-D3, D2-D3, D4-D5, and D2-D5. The D1-D3 variant exhibited rapid association similar to D1 and an initially rapid dissociation that subsequently slowed (*Figure 6C*), suggesting that D2-D3 either indirectly altered D1 binding to dIgA or contributed additional specific interactions. D2-D3 gave inconsistent results among replicate experiments, with weak binding detected in two experiments and no binding detected in a third experiment (data not shown). We did not detect binding of D4-D5 and D2-D5 even at concentrations above 32 μM (data not shown), supporting reports that D1 is required for binding of SC to dIgA (*Brandtzaeg, 2013*).

Differences in binding of hSC D1-D3 versus D1-D5 to dIgA suggested that domains D4 and/or D5 contribute to non-covalent interactions between hSC and dIgA. Hypothesizing that residues near D5 Cys 468/Cys502 are involved, we engineered a series of chimeric proteins (hSC-tCDR1, hSC-tDE and hSC-tCDR1/tDE), in which the human D5 domain contained the CDR1 loop and/or DE loop from counterpart regions of tD2 exhibiting divergent sequence and structure (*Figures 5,6D*). We characterized chimeric protein binding to immobilized dIgA by SPR (*Figure 6E–G*) and calculated equilibrium binding affinities ($K_D$) from maximal binding response values (*Figure 6H*). Binding of hSC ($K_D$~63+/-4nM) and the hSC-tDE chimera ($K_D$~65+/-3nM) were indistinguishable. By contrast, the hSC-tCDR1 chimera exhibited subtle changes during dissociation from dIgA, and the affinity ($K_D$~82 +/-1nM) was slightly reduced compared with the hSC D1-D5 affinity. The affinity for dIgA was further reduced ($K_D$~140+/-7nM), and the dissociation phase was shorter when the binding of a double chimera (hSC-tCDR1/tDE) was evaluated (asterisk, *Figure 6G*). Taken together, these data suggested that D5 mediates direct non-covalent interactions between hSC and dIgA. Since the double chimera exhibited a greater reduction in binding than the hSC-tCDR1 chimeric protein, the D5 DE loop may play a role in stabilizing the position of D5 CDR1, which could be enhanced in the presence of the disulfide that normally links the two motifs (absent in our binding studies). The hSC-tCDR1/tDE did not mimic the binding kinetics of D1 for dIgA, suggesting that additional, as yet unidentified, interfaces contributed to binding affinity and/or that the binding mechanism of D1 was indirectly altered by the presence of D2-D5, perhaps because isolated D1 would not require a conformational change to maximally expose surface area involved in binding. Alternatively, fish motifs found in the chimeric proteins could confer positive binding interactions for dIgA.

Having shown that hSC binds dIgA non-covalently using its D1 and D5 domains and undergoes a conformational change that separates these domains upon SIgA formation (*Figure 4G*, *6E–H*), we characterized binding of shorter hSC proteins, in which the distance between D1 and D5 was constrained. For these studies, we created two hSC proteins, one comprising domains D1-D3-D4-D5 (analogous to avian, reptilian, and amphibian SC, which lack a domain homologous to D2) and another comprising D1-D4-D5 (analogous to a naturally-occurring mammalian splice variant) (*Akula et al., 2014*; *Deitcher et al., 1986*). Both variants exhibited robust, concentration-dependent binding to dIgA; however, dissociation occurred rapidly, with all of the injected protein dissociating ~200 s into the experiment compared to ~400 s for hSC (*Figure 6A,I–J*), similar to the binding kinetics of the D1–dIgA interaction (*Figure 6B*). These results suggest that D2, and possibly D3, contribute to binding by providing direct interactions with ligand and/or by promoting interactions between D5 and dIgA. The latter possibility is attractive, given that chimeric SC with mutations in D5 exhibited similar binding profiles that were characterized by rapid dissociation (*Figure 6G–H*).

We also evaluated the interactions of the hSC variants and chimeras with pIgM and found that D1-D5, D1, D1-D3, D1-D3-D4-D5, and D1-D4-D5 exhibited concentration-dependent binding to pIgM (*Figure 6—figure supplement 1A–C,H,I*). Although qualitative differences in binding among these variants was subtle, the association phase of D1 required a lower starting concentration (16nM) to reach equilibrium compared with the starting concentration required for D1-D5 (256nM), suggesting differences in binding (*Figure 6—figure supplement 1A,B*). Similar to dIgA binding studies, we detected inconsistent responses for D2-D3 and did not detect binding of D4-D5 and D2-D5 (data not shown). In contrast to the dIgA binding studies, we could not detect differences in

pIgM binding between hSC D1-D5 and the hSC-tCDR1, hSC-tDE or hSC-tCDR1/tDE chimeras, suggesting that the D5 CDR1 and DE loop do not play a prominent role in the hSC-pIgM interaction (*Figure 6—figure supplement 1D–G*).

## Discussion

Since the discovery of SC (*Tomasi et al., 1965*) and later identification of its membrane-bound form (*Mostov et al., 1980*), the pIgR has been established as a central component of the vertebrate immune response, transporting and stabilizing secretory antibodies, excluding pathogenic bacteria, providing immune protection at epithelial barriers such as the lungs, gut, urogenital tract, and conferring protection to offspring through breast milk (*Kaetzel, 2005*). Despite its importance, relatively little was known of pIgR/SC structure. Models ranged from schematic representations of elongated tandemly-arranged domains (*Monteiro and Van De Winkel, 2003*) to solution scattering-based, computational models of J-shaped molecules, in which D1 makes contacts only with D2 (via the D1-D2 linker), and D2-D3 are folded back toward D4-D5, leaving all D1 binding motifs free to interact with ligand (*Almogren et al., 2009*; *Bonner et al., 2007*). Here we present atomic resolution structures that detail how the SC structure has changed over the course of vertebrate evolution, showing that bony fish utilize the two domains of ancestral SC to form an open, elongated structure whereas mammals utilize five domains to form a closed, triangular structure that opens upon ligand binding. These differences are accompanied by domain-specific structural variations and together suggest that SC has evolved into a structurally-plastic molecule that protects mammals by specifically modulating its conformation for its known functions as free SC, SIgA, and SIgM. The mammalian SC might be described as the host immune system's hand, whose thumb (D1) and four fingers (D2-D5) can form a fist or open to grasp a polymeric antibody in order to fight potential invaders. By contrast, the more primitive two-domain fish SC could be described as a single finger, already in an open conformation for ligand binding and lacking the dexterity of a hand.

The expansion of SC from two domains to five parallels increasing organism complexity as well as changes in antibody isotypes and species-specific organization of polymeric Igs (*Akula et al., 2014*; *Flajnik, 2010*). While parallel evolutionary changes in receptors and their ligands are common, it was unexpected to find that the addition of D2, D3 and D4 domains in mammals was associated with a closed conformation that occluded putative ligand-binding motifs. Burial of ligand-binding motifs hints that the closed conformation provides mammalian SC with advantages that are separate from its binding to polymeric antibodies. For unliganded SC released into the mucosa, the closed conformation could protect SC from proteases by burying susceptible inter-domain linkers and loops, and also promote interactions with commensal and/or pathogenic bacteria. Supporting the notion that the closed conformation is important for unliganded hSC function, our structure suggests that the D1-D5 interface would be stabilized in acidic mucosa because protonation of D1 His32 that occurs at acidic pH would facilitate electrostatic interactions with conserved acidic residues in D5 (*Figure 3B*). The hSC structure also provides insight into hSC interactions with the human pathogen *Streptococcus pneumoniae*. Unliganded hSC and SIgA use residues in D3 and D4 to bind to the major *S. pneumoniae* adhesion protein CbpA, and a peptide corresponding to hSC D4 residues 349–375 inhibited *S. pneumoniae* adherence to epithelial cells (*Kaetzel, 2005*). The hSC structure shows that residues 349–375 occupy solvent-exposed regions of D4 CDR1 and the D3-contacting regions of the C-C' loop, rationalizing why both D3 and D4 are required for the interaction with CbpA (*Figure 7*). In addition, since SIgA binds CbpA, this suggests that these regions of hSC remain exposed upon binding to dIgA.

Our data support an accepted model for mammalian pIgR binding to dIgA, in which initial non-covalent binding of SC D1 is followed by covalent binding of SC D5 (*Hamburger et al., 2006*), but further suggest that additional steps involving all five domains bridge these two events. Of particular significance is the change in the positions of D1 and D5 upon binding to both dIgA and pIgM, as identified in DEER experiments. The large magnitude of this change (more than 40Å), along with evidence for flexible inter-domain linkers, implies that the D1-D5 interface is broken during ligand binding and suggests that the conformational change involves other domains and exposes additional ligand binding motifs on SC (e.g., by enhancing accessibility to D1 binding motifs). Our SPR data demonstrated that during this process, all five domains contribute to binding kinetics and that D5 binds dIgA independent of covalent bond formation. These observations suggest that non-covalent

interactions between D5 CDR1 and DE loops can stabilize the complex during the disulfide exchange reaction and also suggest that both D1 and D5 contact dIgA directly in endogenous SIgA that lack covalent interactions with SC (*Almogren et al., 2007*; *Lindh and Bjork, 1976*) and in the R1-labeled SC-dIgA complexes used for DEER measurements. In the case of SC binding to IgM, which does not involve covalent bond formation (*Hamburger et al., 2006*), we found that binding also involves a conformational change, which could facilitate ligand binding at secondary sites on SC D1-D5, although such sites do not appear to involve the D5 CDR and DE loops because chimeric SC and hSC exhibited similar binding profiles. The observation that a conformational change occurs even when binding is dominated by D1 supports the hypotheses that the closed SC conformation occludes D1 binding motifs and that increasing D1 accessibility is an important part of the ligand binding mechanism.

The demonstration that hSC D1-D3-D4-D5 and D1-D4-D5 bound dIgA with kinetics similar to D1 rather than to D1-D5 (*Figure 6*) also sheds light on specific roles for pIgR D2-D4 during pIgR-dIgA recognition. These results suggest that the gain of the D2 domain in mammalian pIgR resulted in enhanced binding to dIgA, perhaps by providing direct contacts and/or by facilitating interactions between D5 and ligand, and further suggest that binding of mammalian splice variants to pIg ligands are mediated primarily through interactions with D1. This possibility is supported by earlier work demonstrating that D2 and D3 were required for covalent binding of D5 to dIgA in other mammals (*Crottet and Corthesy, 1999*; *Solari et al., 1985*), an observation we confirmed with an hSC D1-D4-D5 variant (data not shown).

When unliganded, short pIgR/SC variants are likely to adopt conformations that differ from the closed conformation of unliganded hSC. The tSC structure reveals an elongated conformation for a two-domain variant, and analysis of the hSC structure indicates that steric constraints would prevent mammalian D1-D4-D5 splice variants from adopting a closed conformation with a D1-D4-D5 interface equivalent to that of hSC. This implies that D2-D3 play a role in stabilizing the closed hSC conformation. The presence of D3 may be sufficient to stabilize a closed conformation because analysis of the hSC structure suggests that D1-D3-D4-D5 variants could adopt a closed conformation with a D1-D4-D5 interface equivalent to that of hSC. While the absence of a closed conformation might enhance ligand accessibility to D1 binding motifs, it is unclear if this would be advantageous because D1-D4-D5 dissociates from dIgA rapidly compared to D1-D5 (*Figure 6I*). Mammalian SC splice variants transport dIgA in vivo (*Kuhn et al., 1983*); however, the utility of these variants may be impaired by altered interactions with ligands and/or in any context in which the closed state is advantageous, whether for ligand binding and/or free SC functions.

Structural characterization of hSC and SIg complexes using DEER allowed us to capture hSC in both a compact unliganded state and heterogeneous liganded states, information that we could not obtain using crystallography or single particle electron microscopy (unpublished results). While it is notable that SIgA and SIgM appear heterogeneous in structure, predicting how hSC domains are arranged in these complexes is challenging. hSC could bind a single Fc, or bind across the dIgA dimer and contact two Fcs (or more in the case of pIgM). Putative hSC D1 interaction sites on dIgA include Fcα residues 377, 402–411, 413-414, 440–443, and D5 forms a covalent bond with Fcα residue C311 (*Woof and Russell, 2011*). Interaction sites are maximally separated by ~50Å when measured between Phe443 and Cys311 in monomeric Fcα (*Herr et al., 2003*), and likely separated by more than 50Å when measuring between adjacent Fcs in the dIgA dimer (*Figure 7*). Although dIgA contains four copies of all Fc residues implicated in binding, SC binds dIgA with 1:1 stoichiometry (*Kaetzel, 2005*), suggesting either that dimeric IgA formation induces conformational differences in Fcα that alter the binding site environments and/or that a 1:1 stoichiometry is enforced by hSC interactions with J-chain. Solution scattering models suggested that hSC binds along one side of dIgA in an extended conformation that contacts both Fcs (*Bonner et al., 2009*). These results, taken together with our DEER measurements showing that the D1 and D5 CDR1s could be separated by more than 85Å upon hSC binding to dIgA in a SIgA molecule (45Å between D1 and D5 CDR1s in the closed conformation plus >40Å change in positions of D1 and D5 upon ligand binding), favor a model in which hSC contacts both Fcs in dIgA.

Our results suggest a revised model for pIgR/SC function and ligand binding in mammals (*Figure 7*). In this model, unliganded pIgR at the basolateral membrane remains in the closed state until encountering ligand. The closed state, characterized by partially-buried ligand binding motifs including D1 CDR2 and CDR1 residues His32 and Arg34 (*Coyne et al., 1994*; *Roe et al., 1999*), binds to a

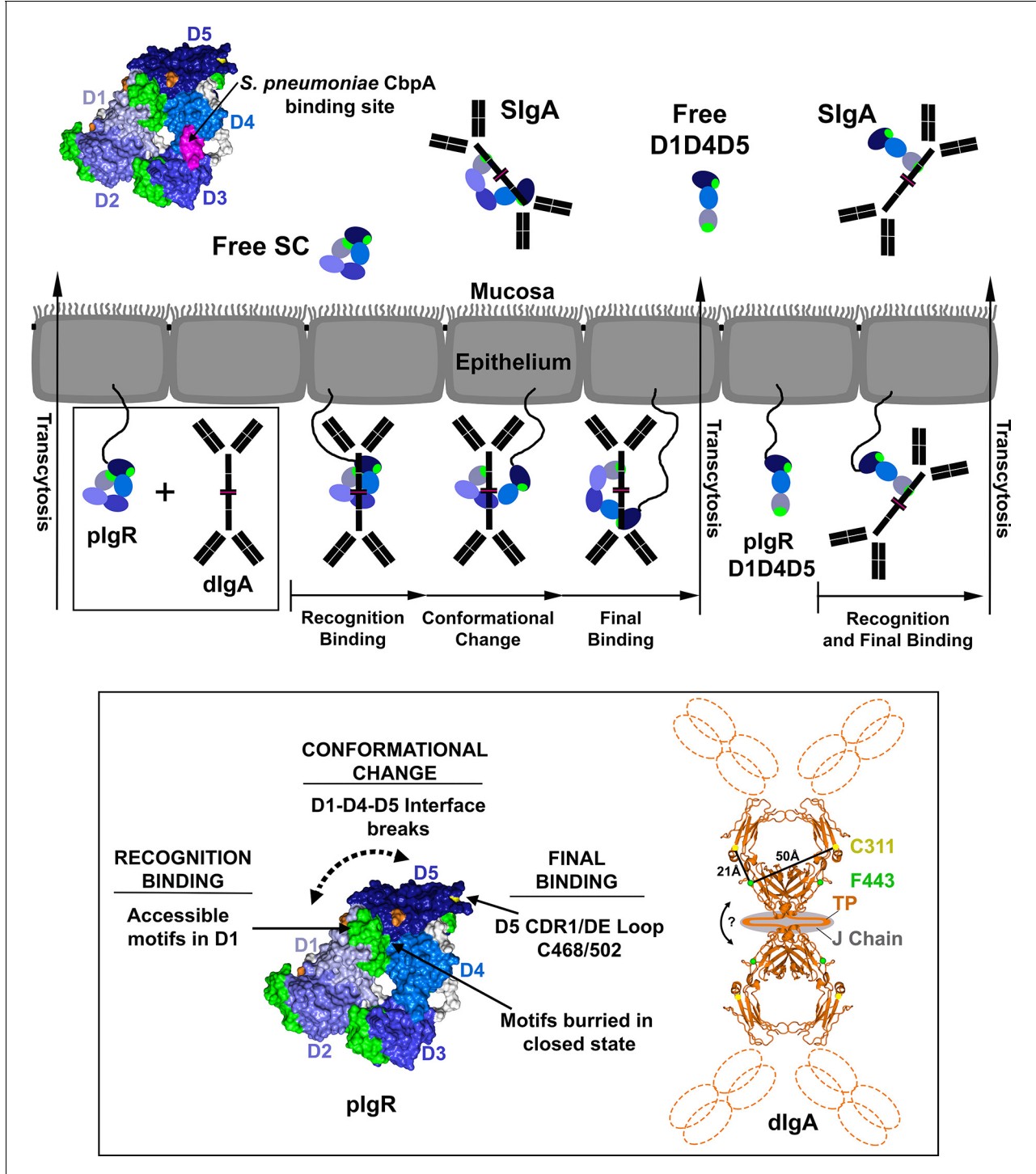

**Figure 7.** Model for pIgR transcytosis, ligand binding and release of free SC and SIgA. Schematic model depicting a mammalian epithelial barrier with membrane-bound SC (pIgR) shown on the basolateral side in its closed, unliganded conformation. The pIgR D1-D5 binding to dIgA is depicted as a three-step mechanism whereas pIgR D1-D4-D5 splice variant binding to dIgA is depicted as a one-step mechanism. pIgR binding to pIgM is not shown but occurs via a mechanism related to dIgA binding. The boxed region (pIgR D1-D5 and dIgA) is enlarged and associated atomic resolution models shown (inset). The hSC crystal structure is colored as in *Figure 1*. A model for dIgA based on the Fcα crystal structure (pdb code 1OW0) is shown with the Fabs, tailpiece (TP) and J chain shown schematically, approximate distances between known SC binding sites labeled (C311 and F443), and an arrow indicating possible bending at the dimer interface. Following transcytosis, free SC and SIgA (and SIgM) are released into the mucosa. Unliganded
*Figure 7 continued on next page*

single Fcα in dIgA (making additional, potential contacts with J-chain) using exposed motifs in D1 CDR1 and CDR3, facilitating an initial recognition event that disrupts the D1-D4-D5 interface and induces a conformational change. The conformational change frees D5 (and perhaps other domains) to form interactions with the second Fc on dIgA (or another Fc on pIgM) and likely exposes D1 residues such as His32 and Arg34, permitting the ligand to interact with regions previously buried by D4-D5. This incremental binding mechanism, particularly the secondary binding at a previously occluded site, might be necessary or advantageous for binding to a ligand in which binding sites are separated by a long distance (e.g., between adjacent Fcs in dIgA or pIgM) and/or to ensure binding at the correct site(s) if multiple binding sites are exposed on a ligand. Following transcytosis to the apical membrane, the pIgR-pIg complexes are released into the mucosa as SIgA or SIgM, and pIg-associated SC retains an open conformation. In contrast, unliganded pIgR released at the apical membrane as free SC would maintain a stable, closed conformation that promotes SC's innate immune function to recognize and exclude bacteria.

In summary, the high-resolution insights on SC structure and mechanism presented here provide a new framework on which to build understanding of mucosal immune evolution and function. Our model for mammalian SC mechanism provides an explanation for how mammalian SC structure evolved to maintain innate functions of unliganded SC while also adapting to bind mammalian pIgs, which are large, complicated molecules. Together with current and forthcoming biological data, these models facilitate insight into normal immune function and disease states while also providing a molecular basis on which to engineer new SC and SIg-based therapeutics.

## Materials and methods

### Construct design

The gene fragment encoding the human pIgR signal peptide and residues 1–547 was amplified from cDNA encoding full-length human pIgR (a gift from Roland Strong, Fred Hutchinson Cancer Research Center) and sub-cloned into the pTT5 expression vector (NRC Biotechnology Research Institute) along with a 5' Kozak sequence and a C-terminal StrepII affinity tag (residues Trp-Ser-His-Pro-Gln-Phe-Glu-Lys). This construct was modified using site-directed mutagenesis to create hSC domain variants, chimeric SC, and mutants used for spin-labeling. In addition to mutating human D5 CDR1 loop residues in the hSC-tCDR1 and hSC-tCDR1/tDE chimeras, we also mutated the flanking residue, Phe466, to a Tyr to preserve stabilizing interactions found in the corresponding fish motifs. To improve purification efficiency, the StrepII tag was replaced with a 6x-His tag for hSC variants produced for spin-labeling and for hSC D1-D3-D4-D5. Residue numbers and sequences of expression constructs are in *Supplementary file 2*.

The gene encoding the *O. mykiss* gene (GenBank: ADB81776.1) was codon-optimized for expression in human cells, synthesized (Blue Heron Biotechnology; Bothell, WA) and cloned into the pTT5 expression vector (NRC Biotechnology Research Institute) with a 5' Kozak sequence. The construct was modified by site directed mutagenesis to insert a C-terminal 6x-His tag and stop codon following residue Ser213 (mature sequence numbering).

### Protein expression and purification

SC proteins and variants were expressed in HEK293-6E cells (NRC Biotechnology Research Institute) transiently transfected with plasmid DNA using 25 kDa linear polyethylenimine (Polysciences; Warrington, PA). Six days following transfection, cell culture supernatants were subjected to affinity chromatography, either StrepTrap or HisTrap (GE Healthcare Bio-Sciences; Pittsburgh,PA), and further purified by SEC using a Superdex 200 column (GE Healthcare). Purifications were conducted in TBS (20 mM Tris-HCl, pH 7.4, 150 mM NaCl) supplemented with either 2.5 mM desthiobiotin or 250 mM imidazole for StrepTrap or HisTrap elution.

Human pIgA was provided by J. Vaerman (Catholic University of Louvain, Brussels, Belgium) (*Vaerman et al., 1995*) and further purified by SEC using a Superose 6 column (GE Healthcare) to isolate dIgA. Human pIgM was purchased from Sigma-Aldrich and further purified by SEC, as described for pIgA, to isolate pentameric IgM.

## Crystallization and data collection

Crystals of Strep-tagged hSC (hSC residues 1–547 plus a Gly-Ser linker) (space group P2₁; $a$ = 61.26 Å, $b$ = 242.43 Å, $c$ = 63.05 Å; β = 114.89°; two molecules per asymmetric unit) were grown by combining 5-7mg/ml protein in TBS with 0.2 M ammonium sulfate, 0.1 M BisTris pH 5.5, 31% Polyethylene glycol (PEG) 3350 using a 1:1 ratio. Crystals grew in sitting drop vapor diffusion at 25°C, and were subsequently soaked in cryoprotectant containing the mother liquor supplemented with 30% glycerol. Iodine derivative crystals of tSC (residues 1–213) were grown by combining 8.2 mg/ml tSC with 75 mM MES monohydrate pH 6.5, 9% w/v PEG 20,000, and 7.5% glycerol in a 1:1 ratio and iteratively soaking resulting crystals in mother liquor supplemented with 10% glycerol/250 mM NaI, 20% glycerol/500 mM NaI, and 30% glycerol/1000 mM NaI for 5, 10, and 60 min respectively. Isomorphous, native crystals (space group P4₃2₁2 (No 96.); $a$ = 54.94 Å, $b$ = 54.94 Å, $c$ = 187.26 Å; one molecule per asymmetric unit) were obtained by combining 8.2 mg/ml tSC with 75 mM MES monohydrate pH 6.5, 9% w/v PEG 20,000, and 25 mM phenol in a 1:1 ratio and were cryoprotected in mother liquor supplemented with 30% glycerol.

Diffraction data from cryopreserved crystals were collected at the Stanford Synchrotron Radiation Laboratory on beamline 12–2 using a PILATUS 6M PAD detector. Data processing was done using Autoxds (A. Gonzoles and Y. Tsai, SSRL), XDS (*Kabsch, 2010*), Pointless, Scala and Truncate as implemented in CCP4i (*Collaborative Computational Project, 1994*).

## Structure determination, and refinement

The crystal structure of hSC was determined using molecular replacement (MR). Search models for each of the five domains were generated by modifying the hSC D1 crystal structure (pdb code 1XED) (*Hamburger et al., 2004*) using Sculptor (*Bunkoczi and Read, 2011*) and performing searches using Phaser (*McCoy et al., 2007*) as implemented in the AutoMR wizard in Phenix (*Adams et al., 2010*). Successful search strategies resulted in placement of eight of ten possible domains in the asymmetric unit, although only D1 was placed in the same position in both hSC copies. Correct domain assignments were determined by inspection of simulated annealing composite omit maps generated in Phenix (*Adams et al., 2010*) and manual building of alternative sequences for D2-D4 using Coot (*Emsley and Cowtan, 2004*). D5, the most divergent domain, was manually built into resulting difference maps using Coot (*Emsley and Cowtan, 2004*). The crystallographic asymmetric unit (ASU) contained two structurally-similar copies of SC (room mean square deviation, rmsd, of 0.29Å for all common Cα atoms).

The final hSC model included residues 2–113, 116–333, 335–426, 432–549 (Chain A), residues 2–256, 260–330, 335–426, 432–549 (Chain B), and N-acetyl glucosamine (NAG) residues attached to Asn65, Asn72, Asn168, and Asn481, an α-1,6 fucose on the Asn72 NAG, and two $SO_4^{2-}$ ions. One $SO_4^{2-}$ is bound to D2 at a position between the A' and B strands; the other is at the interface between D1 and D4, where it contacts the C' and CDR3 from D1 and CDR2, C″ and the C″-D loop in D4. Regions with unresolved main chain density that were not modeled were restricted to the D1-D2 and D3-D4 linkers and the D4 CDR3 loop. The model, including riding hydrogens and water molecules, was refined using Phenix Refine (*Afonine et al., 2012*) using non-crystallographic symmetry (NCS) restraints and individual B-factors and validated using MolProbity (*Chen et al., 2010*). NCS differences are largely restricted to the linkers between domains and flexible loops. For example, the D1-D2 linker residues 111–117, which form crystal contacts with D2 from a neighboring molecule, follows a different path in each NCS copy. This difference causes a 2–3Å shift of Cα atoms in the neighboring D2 C-C' loop as well different side chain conformations of several loop residues.

The structure of tSC was determined by Single Wavelength Anomalous Dispersion (SAD) using iodide ions. Derivative data were collected on SSRL 12–2 operating at 7KeV (1.77Å). Iodide sites were located and phasing was completed using the Shelx CDE pipeline (*Sheldrick, 2010*) as implemented in CCP4 and a preliminary model was built using Phenix Autobuild (*Terwilliger et al., 2008*). A 1.7Å resolution native data set, with the same $R_{free}$-flagged reflections as the Iodide derivative, was used to complete model building and refinement using Coot (*Emsley and Cowtan, 2004*) and Phenix refine (*Afonine et al., 2012*). The structure was validated using MolProbity (*Chen et al., 2010*). The final model included residues tSC 1–52, 60–213 plus four residues of the C-terminal linker/affinity tag.

## Structural analysis, alignments and figures

Structures were analyzed and figures were generated using the Pymol Molecular Graphics System (Schrodinger LLC). Individual domains and NCS copies of hSC were aligned using CEalign in the Pymol Molecular Graphics System (Schrodinger LLC). The hSC domain boundaries (not including linkers) were defined as follows: hD1 residues 2–110, hD2 residues 118–218, hD3 residues 225–329, hD4 residues 339–440 and hD5 residues 446–545 and included sulfate ions and glycans. The tSC domains (not including linker) were defined as tD1 residues 1–52/60–111 and tD2 114–210. Domain interfaces were analyzed using PISA, 'Protein interfaces, surfaces and assemblies' service at the European Bioinformatics Institute. (http://www.ebi.ac.uk/pdbe/prot_int/pistart.html) (*Krissinel and Henrick, 2007*). These analyses utilized modified pdb files in which SC domains were renamed as individual chains that included linkers and were defined as follows: hD1 residues 2–113, hD2 residues 116–222, hD3 residues 223–331, hD4 residues 335–442, hD5 residues 443–549, tD1 1–52, 60–112 and tD2 113–210 and included NAG ligands. Alignments used to generate figures and rmsd calculations for tSC utilized the following residues: tD1 2–6, 7–35, 61–67, 71–91, 102–111 and tD2 114–118, 120–148, 168–174, 177–197, 201–210; tD1 1–23, 29–36, 41–50, 61–93, 103–111 and hD1 3–25, 33–40, 45–54, 63–95, 102–110; tD2 114–115, 119–149, 178–198, 201–210, and hD5 446–447, 450–480, 508–528, 536–545. Inter-domain angles were calculated by determining the angle between the long axes of adjacent domains that had been approximated by ellipsoids calculated from the coordinates using the program Dom_angle (*Su et al., 1998*).

The conformations of CDR loops in the hSC structure do not appear to be heavily influenced by crystal packing because the D1 CDRs are similar in conformation to CDRs in the isolated D1 crystal structure (*Hamburger et al., 2004*) (~0.5–1.0 Å rmsd after aligning D1 Cα atoms) and, crystal contacts involving the CDR1 loops in domains D1–D4 are limited to a single side chain interaction. Several side chains of the D5 CDR1 contact a symmetry-related molecule in the crystal; however, these contacts are unlikely to alter the overall D5 CDR1 conformation because the main chain adopts the same conformation as the D2-D4 CDR1 loops. Crystal contacts with the D2 and D3 CDR2 loops involve extensive interactions that might influence the position of side chains. However, the overall conformations of the short, two-residue CDR2 loop in all SC domains appears more likely constrained by its flanking C'-C'' strands than by crystal contacts, and the D2 and D3 CDR2 loops adopt the same conformation as the CDR2s in other domains, which are not influenced by crystal contacts.

NCBI Genbank accession numbers for SC sequences in alignment figures are: DOG (NP_001274081), COW (NP_776568), MOUSE (NP_035212), RAT (NP_036855), POSSUM (AAD41688) BOAR (NP_999324), CHICKEN (AAQ14493.1), FROG (ABK62772), ANOLE (XP_008113873), *Oncorhynchus mykiss* (ADB81776.1), *Salmo salar* (ACX44838.1), *Epinephelus coioides* (ACV91878.1), *Scophthalmus maximus* (AGN54539.1), *Danio rerio* (NP_001289179.1), *Paralichthys olivaceus* (ADK91435.1), *Cyprinus carpio* (ADB97624.1), *Takifugu rubripes* (NP_001266944.1), *Gadus morhua* (AIR74929.1). Sequence alignments were completed using ClustalOmega (*Sievers et al., 2011*) and corresponding figures were made using Espript 3 (http://espript.ibcp.fr) (*Gouet et al., 1999*; *Gouet et al., 2003*; *Robert and Gouet, 2014*).

## Nitroxide spin-labeling

SEC-purified cysteine-substitution variants of hSC for spin-labeling (T67C/V455C, T67C/Q491C, A80C/V491C, and V455C/Q491C; all including C468A and C502A substitutions) were diluted by 75% in TBS (20 mM Tris-HCl pH 7.4 and 150 mM NaCl), supplemented with 1.5 mM dithiothreitol (DTT) to a final concentration of 0.5 mM DTT, and incubated at 4°C for 20–60 min. DTT was removed using BioSpin P-6 Columns (Bio-Rad; Hercules, CA) in which the protein collection tube contained the required volume of 200 mM 2,2,5,5-tetramethyl-pyrroline-1-oxyl methanethiosulfonate (HO-225) in acetonitrile to produce a five-fold molar excess relative to free cysteine (0.5–1 μl). The HO-225 reagent, which reacts with cysteine to generate the R1 side chain (*Berliner et al., 1982*), was the generous gift of Kalman Hideg (University of Pecs, Hungary). The protein was incubated with HO-225 for 4 hr at 25°C and overnight at 4°C, and then purified by SEC using a Superdex 200 column (GE Healthcare) to remove excess HO-225 and isolate monodisperse protein.

## EPR and DEER spectroscopy

hSC proteins containing R1 were exchanged into TBS supplemented with 20% glycerol and concentrated to 168–220 μM (10–13.5 mg/ml). SIgA and SIgM complexes were formed by mixing either dIgA or pIgM with D1-67R1/D5-491R1 in a 1:1 molar ratio and exchanged into TBS buffer containing 90% $D_2O$ (Sigma-Aldrich; St. Louis, MO) and 20% d-8 glycerol (Sigma-Aldrich) prior to data collection. The final hSC concentration in the samples with dIgA and pIgM was around ~40 μM.

For continuous wave (CW) EPR, samples of 5 μl were loaded into 0.64 mm (inner dimension; i.d.) x 0.84 mm (outer dimension; o.d.) glass capillaries. Spectra were recorded at room temperature on a Varian E-109 spectrometer fitted with a two-loop one-gap resonator (*Hubbell et al., 1987*) at 2 mW incident microwave power and 1 G field modulation amplitude at 100 kHz.

For DEER, samples of 20 μl were loaded into glass capillaries (1.4 i.d. × 1.7 o.d.; VitroCom Inc., NJ) and flash-frozen in liquid nitrogen. Four-pulse DEER data at 80K were obtained on a Bruker ELEXSYS 580 operated at Q-band as previously described (*Lopez et al., 2013*) with some modifications. A standard four-pulse DEER sequence [$(\pi/2)_{vo}$ - $\tau_1$ - $(\pi)_{vo}$ - T - $(\pi)_{vp}$ - $\tau_2$ - $(\pi)_{vo}$ - $\tau_1$ – echo] was employed where $vo$ and $vp$ are the observe and the pump frequencies, respectively. A 36 ns π–pump pulse was set at the maximum of the absorption spectrum and the observer π/2 (16 ns) and π (32 ns) pulses were positioned 50 MHz (17.8 Gauss) upfield at the maximum of the center-field absorption line. The delay time $\tau_1$ was 200 ns; $\tau_2$ varied from 2.5 to 6.0 μs depending on the sample with a constant step size of 16ns. In samples with deuterated buffer, electron spin echo envelope modulation due to the deuterium nuclei was averaged by adding traces at 8 different $\tau_1$ values, starting at 200ns and incrementing by 16ns. Distance distributions were obtained from the raw dipolar evolution data using the program LongDistance available at:

http://www.biochemistry.ucla.edu/biochem/Faculty/Hubbell/. The upper limit of reliable distance ($r$) and width determination ($\sigma$) for each mutant in nanometers was calculated according to (*Jeschke, 2012*):

$$r_{max,\langle r \rangle} \approx 5\sqrt[3]{t_{max}/2\mu s}$$

$$r_{max,\langle \sigma \rangle} \approx 4\sqrt[3]{t_{max}/2\mu s},$$

where $t_{max}$ is the maximum time domain recorded for each sample.

## Surface plasmon resonance

Surface plasmon resonance (SPR) binding studies were performed using a Biacore T200 instrument (GE Healthcare). Human dIgA or pIgM was immobilized on three of four flow cells of CM5 biosensor chips (GE Healthcare) using primary amine coupling (Biacore manual); the remaining flow cell was mock-coupled and used as a reference surface. Ligand densities used for all experiments except those involving D1-D3-D4-D5 were 164, 451, and 765 response units (RUs) for the dIgA surfaces, and 569, 1577, and 4221 RU for pIgM surfaces. Ligand densities used to evaluate D1-D3-D4-D5 binding were 137, 483, and 790 RUs for the dIgA surfaces, and 517, 1779, and 3076 RU for pIgM surfaces. In experiments evaluating D1-D3-D4-D5 binding, D1-D5 and D1 were included as controls to ensure that proteins bound to different sensor chips reproducibly. Sensorgrams shown in figures were generated from the highest density surfaces. A two-fold dilution series in HBS-EP+ buffer (10 mM HEPES pH 7.4, 150 mM NaCl, 3 mM EDTA, 0.05% (v/v) P20) was used to test binding of the following analytes (where the highest concentrations of analyte used on dIgA and pIgM surfaces, respectively, are indicated in parentheses following the analyte name: D1-D5 (1.02 μM, 0.26 μM), D1-D5 C468A/C502A (1.02 μM, 0.26 μM), D1 (1.16 μM, 0.016 μM), D1-D3 (1.02 μM, 0.26 μM), D1-D3-D4-D5 (1.02 μM, 0.26 μM), D1-D4-D5 (1.02 μM, 0.26 μM), D1-D4-D5 C468A/C502A (1.02 μM, 0.26 μM), D2-D3 (65.54 μM, 65.54 μM), D4-D5 C468A/C502A (65.54 μM, 65.54 μM), D2-D5 (32.8 μM, 32.8 μM), CDR1 chimera (1.02 μM, 0.26 μM), DE chimera (1.02 μM, 0.26 μM), CDR1 DE chimera (1.02 μM, 0.26 μM). Additional experiments testing hSC D1-D5 (1.02 μM, 0.26 μM) and D1-67R1/D5-491R1 (1.02 μM, 0.26 μM) binding to dIgA and pIgM utilized identical parameters except that the HBS-EP+ buffer was supplemented with 20% glycerol to mimic conditions used in DEER experiments. Covalent binding of D1-D4-D5 was tested on a single flow cell of a CM5 biosensor

chip with human dIgA surface density of 456 RU. D1-D4-D5 (C468/C502 intact) was manually injected at concentrations up to 4 µM and the total response (RU) was recorded at pre-injection baseline, binding, stability, and following regeneration. The baseline response returned to pre-injection levels following regeneration, indicating that no D1-D4-D5 was covalently bound to the surface. In contrast, the same experiment utilizing 1.02 µM hSC with intact cysteines (C468/C502) resulted in an ~5RU increase in baseline following each injection/regeneration cycle, indicative of covalent binding to the dIgA-coupled surface. In all experiments the flow rate was 50 µl/min, and surfaces were regenerated using 2.5 M $MgCl_2$. All data were collected at 25°C and processed using T200 Evaluation software (GE Heathcare). Equilibrium binding affinity values were calculated using the T200 Evaluation software (GE Healthcare) and the 1:1 Langmuir binding model. The reported affinity values were generated by averaging $K_D$ values obtained from two independent experiments; the corresponding standard deviation was calculated using Excel (Microsoft Corporation). Response data and equilibrium binding models were exported and re-plotted using Prism (GraphPad Software).

## Accession numbers

Crystallographic atomic coordinates and structure factors have been deposited in the Protein Data Bank (http://www.rcsb.org) with codes 5D4K (hSC) and 5F1S (tSC).

## Acknowledgements

We thank Jost Vielmetter and the Caltech Protein Expression Center for assistance with protein expression, and Jens Kaiser, Pavle Nikolovski and the Caltech Molecular Observatory (supported by the Gordon and Betty Moore Foundation) for crystallography support. We also thank members of the Bjorkman and Hubbell labs for insightful discussions, and Collin Kieffer and Anthony West for critical comments on the manuscript.

## Additional information

### Funding

| Funder | Grant reference number | Author |
| --- | --- | --- |
| National Institute of Allergy and Infectious Diseases | AI04123 | Beth M Stadtmueller<br>Kathryn E Huey-Tubman<br>Pamela J Bjorkman |
| Cancer Research Institute | Irving Postdoctoral Fellowship | Beth M Stadtmueller |
| The Jules Stein Professorship Endowment | | Carlos J López<br>Zhongyu Yang<br>Wayne L Hubbell |
| National Institutes of Health | EY005216 | Carlos J López<br>Zhongyu Yang<br>Wayne L Hubbell |

The funders had no role in study design, data collection and interpretation, or the decision to submit the work for publication.

### Author contributions

BMS, CJL, Conception and design, Acquisition of data, Analysis and interpretation of data, Drafting or revising the article; KEH-T, Acquisition of data, Drafting or revising the article; ZY, Acquisition of data, Analysis and interpretation of data; WLH, Analysis and interpretation of data, Drafting or revising the article; PJB, Conception and design, Analysis and interpretation of data, Drafting or revising the article

### Author ORCIDs

Pamela J Bjorkman, http://orcid.org/0000-0002-2277-3990

# Additional files

## Supplementary files

• Supplementary file 1. Crystallographic data collection and refinement statistics. Values in parenthesis refer to the highest-resolution shell. [a]R-meas, redundancy-independent merging R-factor (Diederichs and Karplus, 1997). [b]CC1/2, correlation of one half of the reflections to the other half. [c]CC*, CC1/2 modification showing the correlation of the observed data to unknown true intensities (Karplus and Diederichs, 2012). [d]Rcryst = ($\Sigma$ Fobs-Fcalc)/($\Sigma$ Fobs) and Rfree = Rcryst calculated for 5–10% of reflections from each structure that were excluded from refinement. [e]RMS = Root mean square deviation from ideal value. [f]Determined by Molprobity (*Chen et al., 2010*).

• Supplementary file 2. Expression constructs used in this study. Table listing the name, residue numbers, mutation(s) (endogenous sequence -> mutated sequence) and mature protein sequence for all protein expression constructs used in this study. Residue numbering is based on the mature hSC and tSC sequences and abbreviations used are: SP (Signal peptide), 6HIS (hexahistidine affinity tag), Strep II (Strep II affinity tag), C->A (C468A; C502A mutation).

## Major datasets

The following datasets were generated:

| Author(s) | Year | Dataset title | Dataset URL | Database, license, and accessibility information |
|---|---|---|---|---|
| Stadtmueller BM, Bjorkman PJ | 2016 | Crystal structure of the human polymeric Ig receptor ectodomain | http://www.rcsb.org/pdb/explore/explore.do?structureId=5D4K | Publicly available at the RCSB Protien Data Bank (accession no. 5D4K) |
| Stadtmueller BM, Bjorkman PJ | 2016 | Crystal structure of the teleost fish polymeric Ig receptor ectodomain | http://www.rcsb.org/pdb/explore/explore.do?structureId=5F1S | Publicly available at the RCSB Protien Data Bank (accession no. 5F1S) |

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
