## [Decision Letter]

Thank you for submitting your work entitled "The Structure and Dynamics of Secretory Component and its Interactions with Polymeric Immunoglobulins" for consideration by *eLife*. Your article has been reviewed by three peer reviewers, and the evaluation has been overseen by John Kuriyan as the Senior and Reviewing Editor.

The reviewers have discussed the reviews with one another and the Reviewing editor has drafted this decision to help you prepare a revised submission.

Review:

The manuscript "The Structure and Dynamics of Secretory Component and its Interactions with Polymeric Immunoglobulins" by Stadtmueller et al. presents the crystal structure of the 5 domain human secretory component (hSC) at 2.6 Å resolution. This structure reveals the organization of domains and the location of regions previously implicated in recognition of IgA and IgM, defines folding topology of variable Ig domains, and visualizes domain interfaces and orientations. The authors use DEER spectroscopy to assess solution conformation, heterogeneity and domain flexibility and to reveal a movement of ~40 Å between D1 and D5 domains. These experiments showed that dimeric IgA (dIgA) and polymeric IgM (pIgM) binding caused a conformational change, consistent with the notion that D1 needs to dissociate from the inter-domain interaction with D5 in order to bind ligands. The results from the SPR experiments suggested that Domains 2-5 contribute to ligand binding to various degrees. Based on these results, the authors proposed a model for how pIgR transports dIgA and pIgM and how it may act on its own as an innate immunity receptor.

Next the authors analyze the structure of the teleost fish secretory component. The structure of tSC reveals a protein with IgSF fold, where tD1 and tD2 correspond to D1 and D5, in hSC respectively. They follow this up with a careful dissection of hSC domain affinities to functional ligands and end with a schematic summarizing findings for pIgR transcytosis, ligand binding and release of free SC and SIgA. Lastly the authors investigate the SC-pIg interactions by SPR and show that D1 is required for ligand binding while other domains contribute to it.

The experiments presented in the manuscript are well executed and presented in straightforward fashion. Altogether, the manuscript by Stadtmueller et al. is novel and timely, showing for the first time an intact crystal structure of SC proteins of human and teleost origin.

Major Comments:

The editor and reviewers recognize that the results that are presented in the manuscript represent a substantial body of work. Nevertheless, the following issues were identified by the reviewers as ones that, if addressed, could improve the impact of the findings. Please respond to these comments using your own judgement. We appreciate that additional experimentation may not by feasible in a timely manner, but the manuscript should be revised in the light of these comments:

1) The crystal structures are interesting and represent a substantial advance. The DEER experiments do not contribute much beyond demonstrating that a conformational change occurs upon ligand binding. The actual conformation of the ligand bound SC remains unknown. Similarly, the results on the contributions of different domains to ligand binding are suggestive, but the observed effects are subtle and hard to interpret due to the limited combinations of the domains that were tested. These results do not make it clear how and to what degree each domain is involved in ligand binding, either directly or indirectly through controlling the conformation and accessibility of D1.

2) It is not clear to what extent the inclusion of the fish structure advances understanding. Its inclusion here leaves a number of questions unanswered, such as the degree of conformational reorganization for fish SC function.

3) One particularly interesting aspect indicated by the structure but not followed up by the authors is that the interface between D1 and D5. The authors speculated that this interaction helps protect SC from proteolysis, or stabilizes the conformation that binds commensal and/or pathogenic bacteria. None of these ideas were explored. Moreover, the authors suggest that the D1/D5 interaction may be modulated by pH changes in mucosa, as the interface involves a Histidine residue. This hypothesis is interesting, and could potentially be tested by the DEER assay as shown in Figure 4, through changing the pH of the assay buffer. It might be informative to make some mutations at the interface. Some mutations may enhance binding to dIgA and/or pIgM, and meanwhile disrupt interactions with its bacteria ligands.

---

## [Author Response]

[…] We appreciate that additional experimentation may not by feasible in a timely manner, but the manuscript should be revised in the light of these comments:

1) The crystal structures are interesting and represent a substantial advance. The DEER experiments do not contribute much beyond demonstrating that a conformational change occurs upon ligand binding. The actual conformation of the ligand bound SC remains unknown.

We agree with reviewers that the crystal structures represent a substantial advance and that DEER data clearly demonstrate a conformational change upon ligand binding. In addition, the significance of the DEER data demonstrating that the closed conformation is the predominant solution conformation should not be underestimated. These data show that hSC is closed (until encountering ligand) rather than existing in equilibrium with alternative states such as intermediates on pathway to ligand binding, and validate the crystal structure, eliminating concern that the observed compact conformation resulted from crystal packing. The DEER data further reveal heterogeneous hSC structures upon ligand binding, indicative of binding intermediates and/or complex flexibility. This type of information is important for understanding SIgA/SIgM complex structure(s) and demonstrate that the complex structures are not accessible to methods such as crystallography or electron microscopy that require rigid and uniform structures. Many structural projects would benefit from a similar combined-methods approach, thus we believe our paper will be useful to others studying systems involving flexible protein complexes.

Similarly, the results on the contributions of different domains to ligand binding are suggestive, but the observed effects are subtle and hard to interpret due to the limited combinations of the domains that were tested. These results do not make it clear how and to what degree each domain is involved in ligand binding, either directly or indirectly through controlling the conformation and accessibility of D1.

Interpreting binding data to define precise role(s) for each domain is challenging. This challenge arises largely from the difficulty of identifying accurate kinetic models to describe SPR data as well as the possibility that altering domain arrangement will affect SC conformation in an unpredictable and undetectable manner. While testing additional combinations of domains cannot fully remedy this problem, we have now included new SPR data for the binding of a D1-D3-D4-D5 variant (equivalent to SC from birds, reptiles and amphibians) to dIgA and dIgM. These new data, along with a re-worded Discussion that describes our SPR data in the context of accompanying structural and biophysical data, clarify how individual SC domains participate in the ligand binding mechanism.

2) It is not clear to what extent the inclusion of the fish structure advances understanding. Its inclusion here leaves a number of questions unanswered, such as the degree of conformational reorganization for fish SC function.

The fish SC structure, along with the human SC structure, provides a structural basis for understanding evolutionary changes in mucosal immune function, which include but are not limited to demonstrating that a closed conformation of SC is not required for function in early vertebrates. Furthermore, we used the tSC structure to design the chimeric human/fish pIgR molecules, which are used for binding studies in the manuscript. It would be difficult to rationalize the details of the construction of the chimeric pIgR constructs without presenting the fish SC structure.

3) One particularly interesting aspect indicated by the structure but not followed up by the authors is that the interface between D1 and D5. The authors speculated that this interaction helps protect SC from proteolysis, or stabilizes the conformation that binds commensal and/or pathogenic bacteria. None of these ideas were explored. Moreover, the authors suggest that the D1/D5 interaction may be modulated by pH changes in mucosa, as the interface involves a Histidine residue. This hypothesis is interesting, and could potentially be tested by the DEER assay as shown in Figure 4, through changing the pH of the assay buffer. It might be informative to make some mutations at the interface. Some mutations may enhance binding to dIgA and/or pIgM, and meanwhile disrupt interactions with its bacteria ligands.

Although our data do not fully explore how the closed hSC conformation contributes to specific SC functions, we point out that until now, it was not thought that a closed state of pIgR/SC existed. We anticipate that scientists in other fields will design experiments to address whether the closed SC conformation is important for microbiota homeostasis, whether it prolongs the functional life of SC in the harsh mucosal environment, and whether it promotes the binding of other factors in vivo. However, these questions would require animal studies that are beyond the scope of the present paper.

Determining how pH modulates the stability of the D1-D5 interface is non-trivial. hSC crystallized at a pH of 5.5 and DEER experiments were conducted at a pH of 7.4 and both revealed a closed hSC conformation. Since hSC is closed at basic pH, DEER would not detect enhanced stability of the closed conformation at a lower pH. Within a pH range of 5.5-7.4, the amount of time that hSC spends in the closed conformation in vitro is likely to be constant. However, in the mucosa where hSC may face destabilizing factors, such as age, degradation, and/or interactions with proteins or bacteria, a salt bridge between D1 and D5 may stabilize the closed conformation. Until such destabilizing factors are identified, it would be difficult to test the importance of the histidine-mediated interaction between D1 and D5 in a meaningful way.